# Oral Cavity Squamous Cell Carcinoma Risk Factors: State of the Art

**DOI:** 10.3390/jcm12093264

**Published:** 2023-05-03

**Authors:** Lara Nokovitch, Charles Maquet, Frédéric Crampon, Ihsène Taihi, Lise-Marie Roussel, Rais Obongo, François Virard, Béatrice Fervers, Sophie Deneuve

**Affiliations:** 1Department of Otolaryngology-Head and Neck Surgery, CHU Rouen, 76000 Rouen, France; 2Oral Surgery Department, Rothschild Hospital, 75012 Paris, France; 3URP 2496, Laboratory of Orofacial Pathologies, Imaging and Biotherapies, UFR Odontology, Health Department, Université Paris Cité, 92120 Montrouge, France; 4Department of Head and Neck Cancer and ENT Surgery, Centre Henri Becquerel, 76038 Rouen, France; 5Rouen Cancer Federation, 76000 Rouen, France; 6INSERM U1052-CNRS UMR5286, Cancer Research Center, Centre Léon Bérard, University Claude Bernard Lyon 1, 69008 Lyon, France; 7Faculté d’Odontologie, Hospices Civils de Lyon, University of Lyon, 69002 Lyon, France; 8Département Prévention Cancer Environnement, Centre Léon Bérard, 69008 Lyon, France; 9INSERM UMR 1296, “Radiations: Défense, Santé, Environnement”, Centre Léon Bérard, 69008 Lyon, France; 10Quantification en Imagerie Fonctionnelle-Laboratoire d’Informatique, du Traitement de l’Information et des Systèmes Equipe d’Accueil 4108 (QuantIF-LITIS EA4108), University of Rouen, 76000 Rouen, France

**Keywords:** oral cancer, squamous cell carcinoma, risk factor, occupational exposure, IARC

## Abstract

Head and neck (HN) squamous cell carcinomas (SCCs) originate from the epithelial cells of the mucosal linings of the upper aerodigestive tract, which includes the oral cavity, the pharynx, the larynx, and the sinonasal cavities. There are many associated risk factors, including alcohol drinking coupled with tobacco use, which accounts for 70% to 80% of HNSCCs. Human papilloma virus (HPV) is another independent risk factor for oropharyngeal SCC, but it is only a minor contributor to oral cavity SCC (OSCC). Betel quid chewing is also an established risk factor in southeast Asian countries. However, OSCC, and especially oral tongue cancer, incidence has been reported to be increasing in several countries, suggesting risk factors that have not been identified yet. This review summarizes the established risk factors for oral cavity squamous cell carcinomas and examines other undemonstrated risk factors for HNSCC.

## 1. Introduction

Head and neck (HN) squamous cell carcinomas (SCCs) originate from the epithelial cells of the mucosal linings of the upper aerodigestive tract, which includes the oral cavity, the pharynx, the larynx, and the sinonasal cavities [1]. Alcohol drinking coupled with tobacco use accounts for 70 to 80% of HNSCCs [2]. More recently, human papilloma virus (HPV) has been identified as an independent causative agent for oropharyngeal SCC [3]. Moreover, betel quid chewing is a major risk factor for oral cavity squamous cell carcinoma (OSCC) in many southeast Asian countries, such as India, Sri Lanka, Taiwan, etc. [4,5,6,7,8,9]. 

While the overall incidence of HNSCC continues to decline worldwide, owing to decreased tobacco and alcohol consumption [10,11], oral tongue cancer incidence has been reported to be increasing in several countries [12,13,14]. Interestingly, HPV seems to be involved in only a minor subset of squamous cell carcinomas of the oral cavity [15]. Thus, a rising proportion of patients with HNSCC appear to have neither a significant smoking nor drinking history [16]. This suggests the existence of risk factors not yet identified, or new exposures to established risk factors other than tobacco and alcohol.

This review summarizes the literature about HNSCC risk factors, with an emphasis on established and suspected OSCC risk factors, as well as the role of these factors in patient management and survival.

## 2. Materials and Methods

OSCC was defined according to the International Classification of Diseases for Oncology, third edition (ICD-O-3): C02 to C06. We searched PubMed from 1960 to December 2022, using search terms related to risk factors, etiology, and occupational exposures combined with search terms pertaining to oral cancer sites (Table 1); we further widened the research to HNSCC overall. Human-based epidemiological studies written in English and French were considered. The electronic search was supplemented by hand-searching references from the reference lists of the identified publications. We also searched the International Agency for Research on Cancer (IARC) monographs (http://monographs.iarc.fr/, accessed on 1 March 2023).

Possibly relevant articles were selected through an assessment of titles and abstracts. Studies were selected according to the following criteria: study design (cohort or case–control study); original study or meta-analyses; studies providing information on the association between cancer risk (effect size) and occupational exposure (i.e., included odds ratio [OR], relative risk [RR], standardized incidence ratio [SIR], or mortality rate ratio [SMR]).

When available, we preferentially selected meta-analyses. When an association was consistently supported by numerous studies, the most recent data were retained.

## 3. Results

### 3.1. Established Predisposing Condition for OSCC

Several predisposing conditions have been identified for OSCC, as discussed in the following subsections.

#### 3.1.1. Oral Potentially Malignant Disorders

The oral cavity can present potentially malignant disorders. Premalignant lesions are the most common ones, such as leucoplakia, proliferative verrucous leukoplakia, erythroplasia, oral lichen planus, and oral submucous fibrosis, with different potentials for malignant transformation [17]. The WHO (2005) classifies precancerous lesions according to the level of dysplasia: mild, moderate, severe, and in situ carcinoma [17]. Leukoplakia corresponds to a clinical entity defined, by default, as a “patch or white plaque that cannot be characterized clinically or pathologically as another disease” [17]. This lesion is generally associated with tobacco and alcohol consumption [17]. Oral leukoplakia prevalence worldwide is around 2% [18]. The annual rate of malignant transformation of oral leukoplakia is estimated to be 1% [18]. Risk factors for malignant transformation include the presence of dysplasia, female gender, duration of leukoplakia, location with respect to the tongue or floor of the mouth, occurrence in a non-smoker, size over 2 cm, and non-homogeneous type [18,19].

Proliferative verrucous leukoplakia is a rare entity of unknown cause, characterized mainly by its progressive, aggressive, multifocal, and almost inevitable evolution towards squamous cell carcinoma or verrucous carcinoma [20].

Erythroplasia is a “red velvety lesion that cannot be characterized clinically or pathologically as resulting from another condition” [17]. Surgical resection is recommended due to the higher malignant risk of these lesions compared to leukoplakia, as they are frequently associated with dysplasia and in situ carcinoma [19].

Oral lichen planus is a chronic inflammatory skin and mucous membrane disease that evolves into flare-ups. Its etiology is not clearly elucidated [21]. It manifests itself in several oral clinical forms, of which the three most frequent are reticulated (quiescent), erosive (active), and atrophic (late or post-lichenic) [21]. The malignant transformation of oral lichen planus reported in high-methodological-quality papers was 2.28% [21].

Submucosal fibrosis is a chronic disease of the oral mucosa that affects mainly Asian populations aged 20 to 40 (who consume spicy food, chew Betel quid, etc.) [22]. It is characterized by vesicular stomatitis evolving into erosions with a burning sensation. In the fibrous stage, the mucosa atrophies and loses its elasticity, limiting its function [22]. Cases of malignant transformation have been reported, with a cancer rate in patients with submucosal fibrosis in southeast Asia of about 4.2% [22].

#### 3.1.2. Fanconi Anemia

Fanconi anemia is a hereditary syndrome characterized by bone marrow failure due to a mutational defect in the FANC genes [23]. It is inherited in an autosomal X-linked recessive pattern [23]. FANC genes normally code for proteins that repair damaged DNA and contribute to genomic stability [23]. The risk of HNSCC in patients suffering from Fanconi anemia is 500 to 700 times higher than in the general population, with oral cavity HNSCC being the most frequently observed location [24].

#### 3.1.3. Dyskeratosis Congenita

Dyskeratosis congenita is an hereditary syndrome with dysfunction of telomere maintenance, leading to leukoplakia that can turn into oral cancer [25]. Squamous cell carcinoma is inherited in most cases, and may be X-linked, autosomal dominant, or autosomal recessive, with variable penetrance [26].

#### 3.1.4. Other Genetic Predispositions

Some other genetic determinants of HNSCC have been identified, as recent data suggest that common polymorphisms in DNA repair enzymes, cell-cycle control proteins, apoptotic pathway members, and Fanconi anemia-associated genes likely modulate susceptibility to HNSCC [27].

#### 3.1.5. Familial Aggregation of Cases Apart from Hereditary Syndromes

The French case–control study ICARE including 689 cases of OSCC and 3481 controls, found an increased risk of OSCC for patients with a family history of HNSCC, with an OR of 1.9 CI [1–2.8], after adjusting for tobacco and alcohol consumption [28]. The INHANCE consortium found an increased risk of OSCC in cases of first-degree family history, after adjusting for tobacco and alcohol consumption (OR = 1.53 CI [1.11–2.11]) [29]. The risk increased with the number of family members with OSCC [29]. This risk was even higher in young people below 45 years of age if the history of cancer also affected a young individual (OR = 2.27 CI [1.26–4.1]) [30]. Other studies also reported an increased risk in cases of family history, without adjusting for tobacco and alcohol consumption (data unavailable in this study) [31].

#### 3.1.6. Immune Disorders

Human immunodeficiency virus (HIV) immunodepression is associated with a higher risk of oral cancer in smokers but not in non-smokers, according to a Swiss cohort of 7304 infected patients, with a standardized incidence ratio (SIR) of 4.1, CI [2.1–7.4] [32]. Cancer registry data from New York demonstrated that HIV-infected individuals developing oral cancer were, on average, over a decade younger than non-infected individuals [33]. Immunosuppression related to post-organ transplant treatments was also mentioned as a potential risk factor for HNSCC, especially for those of the oral cavity (SIR = 6.32 CI [3.7–10.1]) in a cohort of 4604 patients [34]. Multisystem chronic graft-versus-host disease (GVHD) was a significant risk factor for oral cancer (RR = 2.9, *p* < 0.001) among patients after 1 year post-transplant [35]. Moreover, oral GVHD induces oral lichenoid lesions that can turn into OSCC [36].

Allogenic hematopoietic cell transplantation is also predisposing to OSCC in a cohort study of 4905 patients, with an SIR of 13.8 CI [11.0–17.2] [37]. In such a population, however, it is difficult to attribute the increased risk of cancer to whole-body irradiation or immunosuppression [37].

### 3.2. Carcinogenic Agents for OSCC Classified by the International Agency for Research on Cancer with Sufficient Evidence

Tobacco and alcohol represent the two main risk factors for HNSCC [38,39]. Other carcinogens are also recognized by the International Agency for Research on Cancer (IARC), with various levels of evidence depending on the HNSCC subsites (Table 2) [40].

#### 3.2.1. Tobacco

Tobacco in all its forms, including smoke, snuff, and chew [41], has been recognized as carcinogenic to humans (Group 1) by the IARC since 1985 [42], with sufficient evidence for the oral cavity [43].

Numerous studies have shown a strong association between tobacco consumption and different HNSCC subsites in men and women. A meta-analysis from 2008 suggests a relative risk (RR) for OSCC of 3.43 CI [2.37–4.94] for smokers and 1.40 CI [0.99–2.00] for former smokers [44].

The risk rises with the number of cigarettes per day and the duration of tobacco consumption, and decreases proportionally to the length of time since cessation [45].

According to the IARC, the tobacco-attributable fraction for oral cavity and pharynx cancers was 71.4% in 2015 (80.2% for men and 41.9% for women) [46].

Passive smoking was recognized as carcinogenic, with limited evidence for the pharynx and larynx [43,47]. A recent case–control study reported that passive smoking is also a risk factor, with sufficient evidence for OSCC [48].

Moreover, except for snuff, all smokeless tobacco products are strongly associated with oral cancer incidence. Shammah showed the highest association OR with 95% confidence intervals (OR = 38.74; CI [19.50–76.96]), followed by oral snuff (OR = 11.80; CI [8.45–16.49), gutka (OR = 8.67; CI [3.59–20.93]), tobacco with betel quid (OR, 7.74; CI [5.38–11.13]), toombak (OR = 4.72; CI [2.88–7.73]), and unspecified chewing tobacco (OR = 4.72; CI [3.13–7.11]) [49].

In addition to being a major risk factor, continuing tobacco smoking represents a major risk factor for recurrence and second primary occurrence, and is associated with poorer survival in HNSCC, including OSCC [50,51,52]. Tobacco cessation after HNSCC diagnosis appears to be an independent predictor of treatment response and long-term survival in a retrospective case–control study from 2022 [53]. Long-term abstinence (>10 pack years) seems to improve significantly overall survival and HNSCC-specific survival in a prospective study from 2023 [54].

#### 3.2.2. Alcohol

Alcohol’s carcinogenic effects derive from the endogenous production of acetaldehyde during its absorption [43]. Acetaldehyde in alcoholic beverages is considered carcinogenic to humans (Group 1) by the IARC [43]. Some populations from eastern Asia presenting mutations of the gene coding for enzymes metabolizing alcohol (alcohol dehydrogenase: ADH) and acetaldehyde (aldehyde dehydrogenase: ALDH) accumulate this molecule and have a higher risk of cancer [55].

A cohort study carried out on more than 490 000 individuals in the United States found a RR of OSCC of 1.52 CI [1.01–2.27] for men consuming more than three glasses of alcohol per day, and a RR of 2.81 CI [1.29–6.11] for women consuming the same amount [56]. A meta-analysis from 2001 finds a meta-RR for a daily intake of more than 100 g of alcohol per day of 6.01 CI [5.46–6.62] for oral cavity and pharynx cancers [57].

Moreover, moderate alcohol consumption (10–20 units/week) is found to increase the risk of mortality independent of other prognostic variables in patients treated for HNSCC [58]. Alcohol use also seems to be an independent predictor of disease-specific survival after treatment for OSCC [59].

#### 3.2.3. Synergistic Effect of Tobacco and Alcohol

Alcohol and tobacco consumption are closely linked in populations [60], probably due to cultural reasons and the tendency of addictive personalities to be more prone to multiplying addictions [61]. Moreover, heavy smokers—with a high number of pack years (PY)—drink more than light smokers with a low number of PY, and heavy drinkers smoke more than light drinkers [62]. In addition, in the case of joint consumption, the effects of alcohol and tobacco are synergistic and multiply instead of adding up [63]. Therefore, if the RR of OSCC is similar for the heavy smoker (RR = 5.8) and drinker (7.4), it tends to multiply these two values with a RR of 37.7 for drinker and smoker patients [64]. Acetaldehyde production is increased by the association of tobacco and alcohol, which might explain the observed synergism [65]. Ethanol might also increase mucous permeability to carcinogens [66]. Indeed, data from an animal model showed that 25% ethanol, alone or associated with nicotine, increased oral cavity mucous membrane permeability to N-nitrosonornicotine (a carcinogen from tobacco) [67]. Thus, in 2009, the fraction attributable to tobacco and alcohol was 64% for OSCC and 72% for pharynx cancer [55].

Smoking and drinking have a detrimental effect on the survival of patients treated for OSCC. In a prospective study including 1165 patients with OSCC, smoking and/or drinking patients had higher all-cause mortality and oral cancer-specific mortality compared to non-smoking and non-drinking patients [68]. This stresses the importance of systematically offering smoking and alcohol cessation to HNSCC patients, including patients with OSCC. Support should be proposed systematically by healthcare professionals concomitant to cancer treatment, especially to patients who are more vulnerable and might need extra support to quit smoking and drinking [69].

#### 3.2.4. Betel Quid and Areca Nut

Betel, which is consumed by chewing, is a mix of leaves from a climbing plant named betel and areca nuts, sometimes mixed with tobacco [4]. Nitrosamines contained in the areca nut mix with saliva and are often responsible for lesions [4]. The IARC has considered betel carcinogenic to humans since 2003 for the oral cavity, with or without tobacco [43]. It is particularly appreciated in India and southeast Asia, with 600 million estimated users in 1999 [70].

A case–control study from 2003 found an OR of 1.83 CI [1.43–2.33] for betel used with tobacco for OSCC [71]. A meta-analysis from 2007 on 11 studies conducted on the effects of betel alone (not associated with tobacco) and adjusted for tobacco consumption reported an OR of 3.50 CI [2.16–5.65] for OSCC [72], whereas the OR associated with tobacco calculated in an Indian case–control study was 5.05 CI [4.26–5.97] for OSCC [71].

#### 3.2.5. HPV

HPV16 has been classified as a Group 1 carcinogen by the IARC since 1995 [73], particularly due to its role in the carcinogenesis of SCC of the cervix uteri.

The evidence has been deemed sufficient for a causal relationship between HPV16 infection and certain cancers of the oral cavity since 1995, whereas the evidence remains limited for HPV18 (Table 3) [73].

Of note, HPV has become, over time, the main cause of HNSCC in the United States, and in northern Europe [74,75]. However, HPV is reported to be responsible for less than 4% of OSCCs [15].

The carcinogenicity of HPV involves several mechanisms, including immortalization of cells, genomic instability, DNA damage repair deficiency, and anti-apoptotic activity [76]. The integration of HPV DNA into the host genome is a key event enabling the expression of viral oncogenes E6 and E7 [76]. E6 protein binds to p53 protein, preventing it from binding to DNA and inhibiting its anti-oncogenic role [76].

Despite their more advanced presentation at diagnosis, HPV-positive HNSCCs, and, more specifically, oropharyngeal squamous cell carcinoma, have improved survival, irrespective of treatment modality [77,78,79]. Yet, their role in OSCC patients remains minor.

### 3.3. Other Potential Risk Factors for OSCC

Apart from the risk factors involved in the occurrence of OSCC with sufficient evidence according to the IARC, and predisposing conditions, some other toxins might also be involved in the occurrence of these cancers. However, the difficulty in investigating OSCC risk factors is that many studies look at the different subsites of HNSCC altogether, and do not allow specific conclusions for the oral cavity. The following sections summarize the available data on suspected risk factors for HNSCC.

#### 3.3.1. Other Addictions

Cannabis consumption is often associated with tobacco, and the illegal nature of its usage makes it difficult to evaluate its role in the development of oral cancer. Its carcinogenic effect in smoked form has not been demonstrated yet for HNSCC; current data report conflicting results and do not support a causal relationship with OSCC [80]. Therefore, its role in the occurrence of HNSCC is controversial [81]. A case–control study by Rosenblatt et al. found no difference between regular cannabis users and controls, with an OR of 0.9 CI [0.6–1.3] [80]. Another study reported an OR of 2.6 CI [1.1–6.6] with a dose-effect response, with an OR of 5 for heavy consumers (more than one cannabis joint per day for 5 years) [82].

In Iran, a case–control study found a significant relationship between opium addiction and OSCC, with an OR of 4 CI [1.2–13.6] [83], and a recent meta-analysis found a four-fold rise in the risk of head and neck cancer among opium users compared to non-users [84].

#### 3.3.2. Mouthwash

Mouthwash usage was suspected to be associated with oral cavity development, as these prepared products contain alcohol in small amounts. Acetaldehyde, the first genotoxic metabolite of ethanol, was found in the saliva of patients regularly using mouthwashes [85]. The ARCAGE study found an OR of 3.23 CI [1.68–6.19] for more than three mouthwashes per day [86]. However, the relationship between mouthwash usage and OSCC is not supported by strong epidemiological evidence [87,88].

#### 3.3.3. Air Pollution

The IARC considers that the current evidence is insufficient to incriminate air pollution in the occurrence of any sub-localization of HNSCC [89,90].

#### 3.3.4. Endocrine Factors

Due to the high proportion of women in patients with OSCC without identified risk factors, an endocrine hypothesis has been suggested [16]. The results of a recent cohort study suggest that menopausal hormone therapy increases the risk of OSCC in postmenopausal women, with an oral estrogen hazard ratio (HR) of 1.633 CI [1.35–1.976] for oral estrogen and an HR of 1.633 CI [1.35–1.976] for tibolone [91]. Moreover, a potential association with endocrine disrupting agents such as bisphenol A has been suggested, but is so far not supported by epidemiological evidence [92].

#### 3.3.5. Occupational Exposures

Several occupational exposures are suspected to play a role in the occurrence of HNSCC (Table 4), and cancers of the oral cavity, in particular. Occupations with an increased risk are butchers, machinists, leather workers, textile industry workers, and sugar cane producers [93].

The available studies are difficult to interpret due to a frequent association with tobacco and alcohol consumption, and the unprecise distinction of HNSCC subsites. If an over-representation of occupational and environmental exposures was observed in a prospective cohort of HNSCC [116], no specific agent was significantly associated with non-smokers and non-drinkers apart from asbestosis for larynx cancer.

Asbestosis is the only occupational risk factor with a level of evidence recognized as sufficient by the IARC, and only for the larynx [94,97]. The ICARE study also demonstrated an additive effect for asbestosis exposure with alcohol consumption (RR = 4.75 CI [−4.29–11.12]), and a supra-additive effect for asbestosis exposure with tobacco consumption (RR = 8.50 CI [0.71–23.81]), as well as for asbestosis exposure with joint tobacco and alcohol consumption (RR = 26.57 CI [11.52–67.88]) [117].

### 3.4. Other Considered Individual Risk Factors for HNSCC

#### 3.4.1. Plummer–Vinson Syndrome

The Plummer–Vinson syndrome (or Kelly Patterson), including esophageal webs, iron-deficiency anemia, and gastrointestinal mucosa atrophy, is frequently associated with HNSCC of the retro-cricoid area (i.e., the hypopharynx) [118]. The improvement of iron intakes lowered its incidence considerably [118]. The mechanism underlying the hypopharyngeal cancer in this condition is poorly understood [118].

#### 3.4.2. Infectious Agents Other Than HPV

Epstein–Barr virus (EBV) is an independent risk factor with sufficient evidence for nasopharyngeal cancer, according to the IARC [119]. While viral hypotheses have been investigated, no convincing link between OSCC and HPV, Herpes Simplex Virus, or EBV has been found [120]. Conversely, Herpes Zoster Virus infection may be protective against OSCC in a nationwide population-based matched control study, with an HR of 0.41 CI [0.33–0.50] in patients diagnosed with HZV infection [120].

#### 3.4.3. Dietary Factors

Numerous studies have suggested that a diet low in fruits, carotenoids, or green vegetables is associated with a higher risk of HNSCC [121]. A study from the INHANCE consortium aggregating 14,520 HNSCC of any subsite and 2337 controls from 22 case–control studies found a protective role for the consumption of fruits (OR = 0.52 CI [0.43–0.62]) and vegetables (OR = 0.66 CI [0.49–0.90]). The consumption of red meat (OR = 1.40 CI [1.13–1.74]) and processed meat (OR = 1.37 CI [1.14–1.65]) was associated with an increased risk of HNSCC [122]. Other studies from the INHANCE consortium found a protective role of vitamin E [123], vitamin C [124], carotenoids [125], and folates [126]. The consumption of dairy products also appears to have a protective role in a case–control study including 959 patients with HNSCC and 2877 controls [127].

The data regarding tea are contradictory, and they are sometimes reported as protective [111]. However, hot mate consumption (categorized as Group 2A by the IARC) appears to increase the risk of HNSCC in a recent meta-analysis by Weber Mello et al., with an OR = 2.24 CI [1.74–2.87] [128]. Coffee would also have a protective role in a meta-analysis from 2011, with a meta-risk of 0.64 CI [0.51–0.80] for SCC of the oral cavity [129].

#### 3.4.4. Body Mass Index

A study from the INHANCE consortium, including 17,666 cases and 28,198 controls, demonstrated a reverse association between HNSCC and waist circumference (adjusted OR of 0.91 CI [0.86–0.95] for a 10 cm increase in waist circumference for men; adjusted OR of 0.86 CI [0.79–0.93] for a 10 cm increase in waist circumference for women) [130]. The relationships between body mass index (BMI) and HNSCC are different among studies, and their analysis is complex [130]. Indeed, smoking and drinking patients are more frequently underweight. The BMI modifies alcohol and tobacco effects in the INHANCE consortium study for oral cavity and pharynx cancers, but not for larynx cancer [130]. In another study by Gaudet et al., reflecting data from 1,941,300 participants, including 3760 HNSCC, the waist circumference and the waist–hip ratio were positively associated with the risk of HNSCC, regardless of smoking status, whereas no association with the BMI was observed for never-smokers [131]. Furthermore, an inverse association was found between height and HNSCC risk (adjusted OR per 10 cm height increase = 0.91 CI [0.86–0.95] for men; adjusted OR = 0.86 CI [0.79–0.93] for women) by the INHANCE consortium in a pooled analysis [29].

Moreover, a pooled analysis including 612 cases and 5580 controls reported a decreased risk of OSCC for patients who practice moderate physical activity with an OR of 0.74 (CI [0.56–0.97]) or intensive physical activity with an OR of 0.53 CI [0.32–0.88] [132].

Nutrition management is crucial to improving the clinical outcome and survival of patients with HNSCC [133]. Thus, in a prospective study involving 1395 patients with OSCC, patients with a BMI < 18.5 kg/m^2^ had a poor survival outcome (HR = 1.585 CI [1.207–2.082]) [134]. Compared with patients with better nutritional status, chemotherapy was also significantly associated with poorer overall survival in malnourished OSCC patients [134].

#### 3.4.5. Dental Hygiene

The INHANCE consortium established an association between oral hygiene and OSCC, with a decreased risk being observed for less than five missing teeth, with an OR of 0.78 CI [0.74–0.82], an annual visit to the dentist (OR = 0.82 CI [0.78–0.87]), daily tooth brushing (OR = 0.83 CI [0.79–0.88]), and the absence of gum disease (OR = 0.94 CI [0.89–0.99]) [135]. No relationship was observed with the wearing of dentures [135]. Other authors found consistent results and mentioned the lack of dental hygiene as a potential risk factor for HNSCC [136].

Chronic traumas were also suggested as a possible cause of SCC of the oral cavity in a meta-analysis incriminating dentures [137] (OR = 3.90 CI [2.48–6.13]), and by some authors alleging the argument frequency, with the most common location of OSCC being the free edge of the tongue, a possible trauma site (unlike the dorsum of the tongue) [138]. Chronic inflammation might also play a role [139].

#### 3.4.6. Socio-Economical Aspects

HNSCCs are more likely to affect socially deprived people, both in France [140] and worldwide [141,142]. Socially deprived people present a higher risk of HNSCC after adjusting for lifestyle risks such as smoking and drinking [141,142]. A study of the national register of cancers, FRANCIM [143], showed an increased risk of HNSCC in socially deprived populations by attributing to each case of HNSCC a social level measured by the European Deprivation Index (EDI) according to the place of residence. Perceptual occupational psychosocial status (SIOPS) appears to be the strongest socioeconomic factor relative to socioeconomic position and manual/non-manual work in a recent pooled case–control study of the INHANCE consortium [144].

## 4. Conclusions

To conclude, tobacco and alcohol consumption remain the main risk factors for OSCC in Europe. Betel quid chewing is also an established risk in southeast Asian countries. Since the 1950s, many countries have campaigned against tobacco and alcohol. The decrease in the prevalence of tobacco and alcohol consumption should result in a decrease in the incidence of HNSCC, and, in a more comprehensive manner, in the incidence of non-HPV-related HNSCC. However, a rise in OSCC incidence is observed. Other risk factors might emerge in the upcoming years and explain the reported diverging incidence trends of oral cancer, especially those regarding oral tongue cancer reported worldwide. For now, the dominant effects of tobacco smoking and alcohol drinking overshadow other potential risk factors. In addition, it is possible that the development of oral cancer in non-smoking, non-drinking patients may require a combination of two conditions or exposures, making their identification complex. Further studies are needed to better understand the observed increasing incidence of oral cancer in non-smoking and non-drinking patients.

Moreover, in patients with OSCC, smoking and alcohol consumption are associated with poorer survival, stressing the importance of systemically offering support for alcohol and tobacco cessation. The increase in OSCC incidence is further leading to more oral cancer survivors requiring follow-up. This is all the more important given the more frequent local relapse, especially in non-smokers and non-drinkers with OSCC.

## Figures and Tables

**Table 1 jcm-12-03264-t001:** Search strategy for exploring oral cancer and HNSCC risk factors or occupational exposures.

Localization/Topic	Search Strategy	Number of References Identified
Cancer of the oral cavity #1	(oral [Tiab] OR mouth [Tiab] OR gingiva [Tiab] OR gingival [Tiab] OR tongue [Tiab] OR palate [Tiab] OR palatal [Tiab] OR buccal [Tiab]) AND (cancer * [Tiab] OR carcinoma * [Tiab] OR malignan * [Tiab] OR tumor * [Tiab] OR tumor * [Tiab] OR neoplasm * [Tiab])	136,837
Etiology #2	(risk * [Title/Abstract] OR risk factor * [Title/Abstract] OR exposure * [Title/Abstract] OR exposed [Title/Abstract])	3,888,254
Combination	#1 AND #2	31,350

**Table 2 jcm-12-03264-t002:** Carcinogenic agents for HNSCC recognized by the IARC with sufficient evidence in humans [40].

	Carcinogenic Agents with Sufficient Evidence in Humans	Carcinogenic Agents with Limited Evidence
Oral cavity	Alcoholic beverages Betel with tobacco Betel without tobacco HPV 16 Smoking Passive smoking	HPV 18 Bitumen, occupational exposure to oxidized and hard bitumen and their emissions during roofing and mastic asphalt work
Oropharynx	HPV 16	
Pharynx	Alcoholic beverages Betel with tobacco HPV 16 Smoking	Asbestosis Opium Printing processes Passive smoking
Larynx	Acid smokes Strong inorganics Alcoholic beverages Asbestosis Opium Smoking	HPV 16 Rubber production Mustard gas Passive smoking

**Table 3 jcm-12-03264-t003:** Classification of HPV types according to their level of carcinogenicity established by the IARC [40].

	Agent is Carcinogenic to Humans, Group 1	Probably Carcinogenic to Humans, Group 2A	Potentially Carcinogenic to Humans, Group 2B
Serotype HPV	16, 18, 31, 33, 35, 39, 45, 51, 52, 56, 58, and 59	68	26, 30, 34, 53, 66, 69, 67, 68, 70, 73, 82, 85, and 97

**Table 4 jcm-12-03264-t004:** Occupational and environmental risk factors, classification of human carcinogenicity according to the IARC, and level of accountability for the four main locations of HNSCC (OR = odds ratio; confidence interval 95% (CI) = []). We distinguished for each cancer site, occupational exposures classified by IARC with “sufficient evidence” (i.e., a causal relationship has been established between exposure to the agent and the given cancer type), “limited evidence” (i.e., a positive association for which a causal interpretation is considered credible, but chance, bias, or confounding could not be ruled out with reasonable confidence), and “inadequate evidence” (i.e., there are no data available in humans, or the available studies are of insufficient quality, consistency, or statistical precision.

		Cancer Site			
Occupational Exposures	Type of Evidence	Oral Cavity	Oropharynx	Hypopharynx	Larynx
Asbestosis IARC classification: Group 1	IARC classification by cancer site	Inadequate evidence	Limited evidence		Sufficient evidence
	Literature	Langevin et al., 2013 [94]: Case-control study OR = 1.41 [1.01 to 1.97]. Paget-Bailly et al., 2012 [55]: Meta-analysis meta-RR = 1.25 [1.10–1.42] (oral cavity and pharynx altogether). Marchand et al., 2000 [95]: Case-control study OR = 1.80 [1.08–2.99] (hypopharynx).			Goodman et al., 1999 [96]: Meta-analysis Meta-RR = 133 [114–155]. Hall et al., 2020 [97]: Case control-study with exposure measurement Cumulative exposure >90th percentile OR = 1.3 [1.0–1.6]
Polycyclic aromatic hydrocarbons IARC classification: Group 1	IARC classification by cancer site	Inadequate evidence			
	Literature	(Oral cavity and pharynx altogether) Paget-Bailly et al., 2012 [55]: Meta-analysis meta-RR = 1.14 [1.02–1.28]			Paget-Bailly et al., 2012 [98]: Meta-analysis Meta-RR = 129 [1.10–1.52]
Wood dusts IARC classification: Group 1	IARC classification by cancer site	Inadequate evidence			Limited evidence
	Literature	Smailyte et al., 2012 [99]: Cohort study SIR = 2.83 [1.29–5.37]			Langevin et al., 2013 [94]: Case control study OR = 1.2 [1.0–1.3]
Metal Dust IARC classification: Group 2B (industry)	IARC classification by cancer site	Inadequate evidence			Limited evidence
	Literature				Langevin et al., 2013 [94] Case control study OR = 1.2 [1.0–1.4]
Textile industry IARC classification: Group 2B (industry)	IARC classification by cancer site	Inadequate evidence			
	Literature				Paget-Bailly et al., 2012 [94]: Meta-analysis meta-RR = 1.41 [1.09–1.53] (textile dust) (Paget-Bailly et al., 2012 [55])
Industrial rubber manufacturing process IARC classification: Group 1	IARC classification by cancer site	Inadequate evidence			Limited evidence
	Literature				Paget-Bailly et al., 2012 [98]: Meta-analysis meta-RR= 1.39 [1.13–1.71]
Silica dust IARC classification: Group 1	IARC classification by cancer site	Inadequate evidence			
	Literature				Chen et al., 2012 [100]: Meta-analysis OR = 1.39 [1.17–1.67]. Hall et al., 2020 [97]: Case-control study with exposure measurement OR = 1.4 [1.2–1.7] Cumulative exposure 75th–90th percentile OR = 1.4 [1.1–1.8].
Diesel exhaust IARC classification: Group 1	IARC classification by cancer site	Inadequate evidence			
	Literature				Paget-Bailly et al., 2012 [98]: Meta-analysis meta-RR = 1.16 [1.10–1.33]
Uranium IARC classification: Group 1	IARC classification by cancer site	Inadequate evidence			
	Literature				Dupree et al., 1987 [101]: Cohort study SMR = 447 [144–1043]
Mustard gas IARC classification: Group 1	IARC classification by cancer site	Inadequate evidence		Limited evidence	
	Literature		Easton et al., 1988: Cohort study [102] Standardized mortality ratio (SMR) = 549	Easton et al., 1988 [102]: Cohort study SMR = 273	
Occupational Exposure risk Factors		Oral Cavity	Oropharynx	Hypopharynx	Larynx
Chlorinated solvents IARC classification: Group 1 Trichloroethylene (TCE) Group 2A Perchloroethylene (PCE)	IARC classification by cancer site	Inadequate evidence			
	Literature		Carton et al., 2017 [103]: Case-control study PCE: OR = 2.97 [1.05–8.45]; trichloroethylene TCE: OR = 2.15 [1.21–3.81]		
					Barul et al., 2017 [104]: Case-control study PCE OR = 3.86 [1.30–11.48]
Welding fumes IARC classification: Group 1	IARC classification by cancer site	Inadequate evidence			
	Literature			Gustavsson et al., 1998 [105]: Case-control study RR = 2.3 [1.10–4.7]	Gustavsson et al., 1998 [105]: Case-control study RR = 2.0 [1.0–3.7]
		Barul et al., 2020 [106]: Case-control study OR = 1.31 [1.03–1.67]			
Domestic emissions from domestic coal combustion IARC classification: Group 1	IARC classification by cancer site			No evidence	
	Literature			Shangina et al., 2006 [107]: Case-control study OR = 4.19 [1.18–14.89]	
				Sapkota et al., 2008 [108]: Coal period of use > 50 years hypopharynx OR = 3.47 [0.95–12.69] and larynx OR = 3.65 [1.11–11.93].	
Vegetable dusts No IARC classification	IARC classification by cancer site	Inadequate evidence			
	Literature				Laakkonen et al., 2006 [109]: Cohort study SIR = 3.55 [1.3–7.72]
Occupational Exposure Risk Factors		Oral Cavity	Oropharynx	Hypopharynx	Larynx
Flour dust IARC classification: Group 3	IARC classification by cancer site		No evidence		
	Literature		Carton et al., 2018 [110]: Case control study OR = 1.55 [1.11–2.17]		
Leather dust IARC classification: Group 1	IARC classification by cancer site	Inadequate evidence			
	Literature	Radoï et al., 2019 [111]: Case-control study OR = 2.26 [1.07–4.76]			
Mustard gas IARC classification: Group 1	IARC classification by cancer site	Limited evidence	No evidence	Limited evidence	
	Literature	Easton et al., 1988 [102]: Cohort study SMR = 280	Easton et al., 1988 [102]: Cohort study SMR = 549	Easton et al., 1988 [102]: Cohort study SMR = 273	
Bitumen IARC classification: Oxidized bitumen and its emissions during roofing ○ Group 2B Straight run bitumen and its emissions during road paving ○ Group 2B Hard bitumen and its emissions during mastic asphalt work ○ Group 2B	IARC classification by cancer site	Limited evidence			
	Literature	Hansen, 1989 [112] Cohort study SMR= 1111 [135–4014]	Mundt et al., 2018 [113] Meta-analysis meta-RR = 1.31 [1.03–1.67]		
Printing process IARC classification: Group 1	IARC classification by cancer site	Limited evidence			
	Literature	Lloyd et al., 1977 [114]: Cohort study RR =2.4 [1.2–4.6]. Leon et al., 1994 [115]: Cohort study editors SMR = 1053 [128–3803] office workers SMR = 638 [132–1864].			

## Data Availability

The data are available upon request from the corresponding author.

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
