# Peer review of "Oral Cavity Squamous Cell Carcinoma Risk Factors: State of the Art"

_jcm, 2023, doi:10.3390/jcm12093264_

Round 1

Reviewer 1 Report

The methodology is adequate but the incorporation of the PRISMA flow diagram in the methodology is suggested.

There is a suggestion regarding the use of one of the popularly used term ‘oral potentially malignant disorders’ at the proper place in the discussion. 

The authors have described one of the important topic. As the prevalence of OSCC is increasing, to study the risk factors for the same is definitely of great concern. Introduction is relevant, adequate and properly written.

Results – properly mentioned

Sufficient discussion is included depicting the importance of the findings and their relevance to future understanding of the risk factors for OSCC.

The overall paper is interesting and useful to readers

Author Response

The methodology is adequate but the incorporation of the PRISMA flow diagram in the methodology is suggested.

Answer: The PRISMA statement and flowchart have been developed to provide guidance in systematic reviews and meta-analyses. Systematic reviews are the top level of the evidence pyramid and have gained importance to synthesize evidence that is reasonably consistent across the individual studies. Yet, for some review topics the narrow focus of systematic reviews constitutes an important limitation. Tobacco and alcohol consumption remain the predominant risk factors in of oral cavity SCC and their increasing incidence in HPV negative patients without tobacco and alcohol consumption, has only very recently gained awareness, and these rare clinical situations remain poorly identified in individual etiological studies. Therefore the narrative review of the present manuscript does allow for a more comprehensive coverage of the literature including IARC monographs and published reviews covering a wider range of localizations and risk factors, and thus provide more potential for insight into potential risk factors.

There is a suggestion regarding the use of one of the popularly used term ‘oral potentially malignant disorders’ at the proper place in the discussion. 

Answer: Thank you for this suggestion. We preferred to use the term ‘risk factor’ in the conclusion rather than ‘oral potentially malignant disorders’ that would have been too restrictive. Indeed, the scope of OSCC risk factors is not limited only to oral mucosal disorders with increased risk for malignant transformation. However, we took into account your suggestion and  replaced the term ‘precancerous lesion’ by ‘oral potentially malignant disorders’ in the result section at the proper place.

The authors have described one of the important topic. As the prevalence of OSCC is increasing, to study the risk factors for the same is definitely of great concern. Introduction is relevant, adequate and properly written.

Results – properly mentioned

Sufficient discussion is included depicting the importance of the findings and their relevance to future understanding of the risk factors for OSCC.

The overall paper is interesting and useful to readers

Reviewer 2 Report

This is a potential interesting manuscript dealing the risk factors of oral squamous cell carcinoma. Some revisions (based on comments) are necessary to improve the data presentation

1.     Since betel quid chewing is also one main etiologic factor of oral cancer in some countries. Betel quid is better to be added into “Abstract”.

2.     After line 37, please add “Moreover, betel quid chewing is one major risk factor of OSCC in many southeast Asian countries such as India, Sri Lanka, Taiwan etc. (Int J Cancer. 2014 Sep 15;135(6):1433-43.; Oral Oncol. 2001 Sep;37(6):477-92; Oral Oncol. 2010 Apr;46(4):297-301; Int J Mol Sci. 2020 Oct 30;21(21):8104; Asian Pac J Cancer Prev. 2018 Sep 26;19(9):2485-2492; Br J Psychiatry. 2012 Nov;201(5):383-91.)”.

3.     Line 43-44: the meaning of sentences is not clear. The existence of additional risk factors such as….?

4.     Line 80: age of leukoplakia?

5.     Line 98: …of varying rate ranging from ? to ? (references)

6.     Leukoplakia can be used instead of leucoplakia through the text?

7.     3.2.4, title can be changed as “betel quid and areca nut”. Betel quid (as mixture) and areca nut (nut only).

8.     Line 158-170, line 197-198, line 213-225: Some small paragraphs can be merged. Please check through text.

Author Response

This is a potential interesting manuscript dealing the risk factors of oral squamous cell carcinoma. Some revisions (based on comments) are necessary to improve the data presentation

  1. Since betel quid chewing is also one main etiologic factor of oral cancer in some countries. Betel quid is better to be added into “Abstract”.

Answer: Thank you for pointing this out. We added a sentence regarding betel chewing in the abstract as suggested.

  1. After line 37, please add “Moreover, betel quid chewing is one major risk factor of OSCC in many southeast Asian countries such as India, Sri Lanka, Taiwan etc. (Int J Cancer. 2014 Sep 15;135(6):1433-43.; Oral Oncol. 2001 Sep;37(6):477-92; Oral Oncol. 2010 Apr;46(4):297-301; Int J Mol Sci. 2020 Oct 30;21(21):8104; Asian Pac J Cancer Prev. 2018 Sep 26;19(9):2485-2492; Br J Psychiatry. 2012 Nov;201(5):383-91.)”.

Answer : The sentence and references you suggested were added after line 37.

  1. Line 43-44: the meaning of sentences is not clear. The existence of additional risk factors such as….?

Answer: We apologize if this sentence wasn’t clear enough. This paragraph was about OSCC in non smokers non drinkers, suggesting the existence of unknown risk factors or new exposures to carcinogenic agents other than alcohol and tobacco. The sentence was modified accordingly, and we hope you’ll find it clearer.

  1. Line 80: age of leukoplakia?

Answer: The term ‘age’ was probably inadequate. We modified it and used the term ‘duration’ instead.

  1. Line 98: …of varying rate ranging from ? to ? (references)

Answer: Thank you for pointing out this inaccuracy. The malignant transformation rate of submucosal fibrosis was precised, and a reference was added.

  1. Leukoplakia can be used instead of leucoplakia through the text?

Answer: The term leucoplakia was replaced by leukoplakia as suggested through the text.

  1. 2.4, title can be changed as “betel quid and areca nut”. Betel quid (as mixture) and areca nut (nut only).

Answer: The title was changed accordingly to your suggestion.

  1. Line 158-170, line 197-198, line 213-225: Some small paragraphs can be merged. Please check through text.